# ERROR CONTROLLED ACTOR-CRITIC METHOD TO REINFORCEMENT LEARNING

## ABSTRACT

In the reinforcement learning (RL) algorithms which incorporate function approximation methods, the approximation error of value function inevitably causes overestimation phenomenon and impacts algorithm performances. To mitigate the negative effects caused by approximation error, we propose a new actor-critic algorithm called Error Controlled Actor-critic which ensures confining the approximation error in value function. In this paper, we derive an upper boundary of approximation error for Q function approximator in actor-critic methods, and find that the error can be lowered by keep new policy close to the previous one during the training phase of the policy. The results of experiments on a range of continuous control tasks from OpenAI gym suite demonstrate that the proposed actor-critic algorithm apparently reduces the approximation error and significantly outperforms other model-free RL algorithms.

## 1 INTRODUCTION

Reinforcement learning (RL) algorithms are combined with function approximation methods to adapt to the application scenarios whose state spaces are combinatorial, large, or even continuous. Many function approximation methods RL methods, including the Fourier basis (Konidaris et al., 2011), kernel regression (Xu, 2006; Barreto et al., 2011; Bhat et al., 2012), and neural neworks (Barto et al., 1982; Tesauro, 1992; Boyan et al., 1992; Gullapalli, 1992) have been used to learn value functions. In recent years, many deep reinforcement learning (DRL) methods were implemented by incorporating deep learning into RL methods. Deep Q-learning Network (DQN) (Mnih et al., 2013) reported by Mnih in 2013 is a typical work that uses a deep convolutional neural network (CNN) to represent a suitable action value function estimating future rewards (returns); it successfully learned end-to-end control policies for seven Atari 2600 games directly from large state spaces. Thereafter, deep RL methods, such as Deep Deterministic Policy Gradient (DDPG) (Lillicrap et al., 2016), Proximal Policy Optimization (PPO) (Schulman et al., 2017), Twin Delayed Deep Deterministic policy gradient (TD3) (Fujimoto et al., 2018), and Soft Actor-Critic (SAC) (Haarnoja et al., 2018), started to become mainstream in the field of RL.

Althouth function approximation methods have assisted reinforcement learning (RL) algorithms to perform well in complex problems by providing great representation power; however, they also cause an issue called overestimation phenomenon that jeopardize the optimization process of RL algorithms. Thrun & Schwartz (1993) presented a theoretical analysis of this systematic overestimation phenomenon in Q-learning methods that use function approximation methods. Similar problem persists in the actor-critic methods employed function approximation methods. Thomas (2014) reported that several natural actor-critic algorithms use biased estimates of policy gradient to update parameters when using function approximation to approximate the action value function. Fujimoto et al. (2018) proved that the value estimation in the deterministic policy gradient method also lead to overestimation problem. In brief, the approximation errors of value functions caused the inaccuracy of estimated values, and such inaccuracy induced the overestimation on value function; so that poor performances might be assigned to high reward values. As a result, policies with poor performance might be obtained.

Previous works attempted to find direct strategies to effectively reduce the overestimation. Hasselt (2010) proposed Double Q-learning, in which the samples are divided into two sets to train two independent Q-function estimators. To diminish the overestimation, one Q-function estimator is

used to select actions, and the other one is applied to estimate its value. Fujimoto et al. (2018) proposed mechanisms, including clipped double Q-learning and delayed policy updates, to minimize the overestimation.

In contrast to these methods, we focus on actor-critic setting and manage to reduce the approximation error of value function, which is the source of the overestimation, in an indirect but effective way. We use the concepts of domain adaptation (Ben-David et al., 2010) to derive an upper boundary of the approximation error in Q function approximator. Then, we find that the least upper bound of this error can be obtained by minimizing the Kullback-Leibler divergence (KL divergence) between new policy and its previous one. This means minimizing the KL divergence when traning policy can stabilize the critic and then confine the approximation error in Q function. Interestingly, we arrive at similar conclusion as two literatures Geist et al. (2019); Vieillard et al. (2020) by a somewhat different route. In their works, the authors directly studied the effect of KL and entropy regularization in RL and proved that a KL regularization indeed leads to averaging errors made at each iteration of value function update. While our idea is very different from theirs: It is impracticable to minimize the approximation error directly, so instead we try to minimize an upper bound of approximation error. This is similar to Expectation-Maximization Algorithm (Bishop, 2006) which maximize a lower bound of log-likelihood instead of log-likelihood directly. We derive an upper boundary of approximation error for Q function approximatorin actor-critic methods, and arrive at a more general conclusion: approximation error can be reduced by keep new policy close to the previous one. Note that KL penalty is a effective way, but not the only way.

Furthermore, the mentioned indirect operation (i.e. the KL penalty) can work together with the mentioned direct strategies for reducing overestimation, for example, clipped double Q-learning. Then, a new actor-critic method called Error Controlled Actor-critic (ECAC) is established by adopting an effective operation that minimizes the KL divergence to keep the upper bound as low as possible. In other words, this method ensures the similarity between every two consecutive polices in training process and reduces the optimization difficulty of value function, so that the error in Q function approximators can be decreased. Ablation studies were performed to examine the effectiveness of our proposed strategy for decreasing the approximation error, and comparative evaluations were conducted to verify that our method can outperform other mainstream RL algorithms.

The main contributions of this paper are summarized as follow: (1) We presented an upper boundary of the approximation error in Q function approximator; (2) We proposed a practical actor-critic method—ECAC which decreases the approximation error by restricting the KL divergence between every two consecutive policies and adopt a mechanism to automatically adjust the coefficient of KL term.

## 2 PRELIMINARIES

### 2.1 REINFORCEMENT LEARNING

Reinforcement learning (RL) algorithms are modeled as a mathematical framework called Markov Decision Process (MDP). In each time-step of MDP, an agent generates an action based on current state of its environment, then receives a reward and a new state from the environment. Environmental state and agent's action at time $t$ are denoted as $s_t \in \mathcal{S}$ and $a_t \in \mathcal{A}$, respectively; $\mathcal{S}$ and $\mathcal{A}$ denote the state and action spaces respectively, which may be either discrete or continuous. Environment is described by a *reward function*, $r(s_t, a_t)$, and a *transition probability distribution*, $Pr(s_{t+1} = s'|s_t = s, a_t = a)$. Transition probability distribution specifies the probability that the environment will transition to next state. Initial state distribution is denoted as $Pr_0(s)$.

Let $\pi$ denotes a policy, $\eta(\pi)$ denotes its expected discounted rewards:

$$\eta(\pi) = E_\pi[R_1 + \gamma R_2 + \gamma^2 R_3 + \cdots] = E_\pi[\sum_{t=0}^{\infty} \gamma^k R_{t+1}], \quad (1)$$

where $\gamma$ denotes a discount rate and $0 \leq \gamma \leq 1$. The goal of RL is to find a policy, $\pi^*$, that maximizes a performance function over policy, $J(\pi)$, which measures the performance of policy:

$$\pi^* = arg \max_\pi J(\pi). \quad (2)$$

A natural form of $J(\pi)$ is $\eta(\pi)$. Different interpretations of this optimization goal lead to different routes to the their solutions.

Almost all reinforcement learning algorithms involve estimating value functions, including state-value and action-value functions. State-value function, $V^\pi(s)$, gives the expected sum of discounted reward when starting in $s$ and following a given policy, $\pi$. $V^\pi(s)$ specified by:

$$V^\pi(s) = E_\pi[\sum_{k=0}^\infty \gamma^k R_{t+k+1} | s_t = s]. \tag{3}$$

Similarly, action-value function, $Q^\pi(s, a)$, is given by:

$$Q^\pi(s, a) = E_\pi[\sum_{k=0}^\infty \gamma^k R_{t+k+1} | s_t = s, a_t = a]. \tag{4}$$

## 2.2 ACTOR-CRITIC ARCHITECTURE

To avoid confusion, by default, we discuss only RL methods with function approximation in this section.

RL methods can be roughly divided into three categories: 1) *value-based*, 2) *policy-based*, and 3) *actor-critic* methods. *Value-based* method only learn value functions (state-value or action-value functions), and have the advantage of fast convergence. *Policy-based* methods primarily learn parameterized policies. A parameterized policy (with parameter vector, $\theta$) is either a distribution over actions given a state, $\pi_\theta(a|s)$, or a deterministic function, $a = \pi_\theta(s)$. Their basic update is $\theta_{n+1} = \theta_n + \alpha \nabla J(\theta_n)$ where is learning rate. Policy based methods show better convergence guarantees but have high variance in gradient estimates. *Actor-critic* methods learn both value functions and policies and use value functions to improve policies. In this way, they trade off small bias in gradient estimates to low variance in gradient estimates.

*Actor-critic* architecture (Peters & Schaal, 2008; Degris et al., 2013; Sutton & Barto, 2018) consists of two components: *actor* and *critic* modules. Critic module learns learns state-value function, $V_\phi(s)$, or action-value function, $Q_\phi(s, a)$ or both of them, usually by temporal-difference (TD) methods. Actor module learns a stochastic policy, $\pi_\theta(a|s)$, or a deterministic policy, $a = \pi_\theta(s)$, and utilizes value function to improve the policy. For example, in actor module of DDPG (Lillicrap et al., 2016), the policy is updated by using the following performance function

$$J(\theta) = \mathbb{E}_{\pi_\theta}[Q_\phi(s_t, \pi_\theta(s_t))], \tag{5}$$

where $\pi_\theta(s_t)$ is a deterministic policy.

## 2.3 DOMAIN ADAPTATION

*Domain adaptation* is a task which aims at adapting a well performing model from a source domain to a different target domain. It is used to describe the task of critic module in section 3.2. The learning task of critic module is viewed as adapting a learned Q function approximator to next one, and the target error equates to the approximation error at current iteration of critic update. Here, we present some concepts in domain adaptation, including domain, source error, and target error.

*Domain* is defined as a specific pair consisting of a distribution, $P$, on an input space, $\mathcal{X}$, and a labeling function, $f : \mathcal{X} \to \mathbb{R}$. In domain adaption, source and target domains are denoted as $\langle P_S, f_S \rangle$ and $\langle P_T, f_T \rangle$, respectively. A function, $h : \mathcal{X} \to \mathbb{R}$, is defined as a hypothesis.

*Source error* is the difference between a hypothesis, $h(x)$, and a labeling function of source domain, $f_S(x)$, on a source distribution which is denoted as follow:

$$e_S(h, f_S) = \underset{x \sim P_S}{E} [|h(x) - f_S(x)|]. \tag{6}$$

*Target error* is the difference between a hypothesis, $h(x)$, and a labeling function of target domain, $f_T(x)$, on a target distribution which is denoted as follow:

$$e_T(h, f_T) = \underset{x \sim P_T}{E} [|h(x) - f_T(x)|]. \tag{7}$$

For convenience, we use the shorthand $e_S(h) = e_S(h, f_S)$ and $e_T(h) = e_T(h, f_T)$.

## 3  ERROR CONTROLLED ACTOR-CRITIC

To reduce the impact of the approximation error, we propose a new actor-critic algorithm called error controlled actor-critic (ECAC). We present the details of a version of ECAC for contious control tasks, and, more importantly, explain the rationale for confining the KL divergence. In section 3.2, we will show that, at each iteration of critic update, the target error equals to the approximation error of Q function approximator. Then, we derive an upper boundary of the error, and find that the error can be reduced by limiting the KL divergence between every two consecutive policies. Although this operation is conducted when training the policy, it can indirectly reduce the optimization difficulty of Q function. Moreover, this indirect operation can work together with the strategies for diminishing overestimation phenomenon. We incorporate clipped double-Q learning into the critic module.

### 3.1  CRITIC MODULE–LEARNING ACTION VALUE FUNCTIONS

The learning task of critic module is to approximate Q functions. In the critic module of ECAC, two Q functions are approximated by two neural networks with weight $\phi^{(1)}$ and $\phi^{(2)}$, respectively. As noted previously, we adopt clipped double-Q strategy (Fujimoto et al., 2018) to directly reduce overestimation. Furthermore, we adopt experience replay mechanism (Lin, 1992)—agent's experiences at each timestep, $(\boldsymbol{s}_t, \boldsymbol{a}_t, r_{t+1}, \boldsymbol{s}_{t+1})$, are stored in a replay buffer, $\mathcal{D}$; and training samples are uniformly drawn from this buffer. The two Q networks, $Q_{\phi^{(1)}}$ and $Q_{\phi^{(2)}}$ , are trained by using temporal-difference learning. Notice that the bootstrapping operation in this setting—uses function approximation—means to minimize the following two TD errors of Q funtion:

$$\delta_t^{(j)} = Q_{\phi^{(j)}}(\boldsymbol{s}_t, \boldsymbol{a}_t) - (R_{t+1} + \gamma \min_{i=1,2} Q_{\phi^{(i)}}(\boldsymbol{s}_{t+1}, \boldsymbol{a}_{t+1})). \tag{8}$$

Notice that clipped double-Q learning uses the smaller of the two Q values to form the TD error. With the minimum operator, it decreases the likelihood of overestimation by increasing the likelihood of underestimation. The two Q networks are respectively trained by minimizing the following two loss functions:

$$L(\phi^{(1)}) = \underset{\substack{(\boldsymbol{s}, \boldsymbol{a}, r, \boldsymbol{s}') \sim P_{\mathcal{D}} \\ \boldsymbol{s}' \sim Pr(\cdot | \boldsymbol{s}, \boldsymbol{a}), \\ \boldsymbol{a}' \sim \pi(\cdot | \boldsymbol{s}')}}{\mathrm{E}} [Q_{\phi^{(1)}}(\boldsymbol{s}, \boldsymbol{a}) - (r + \gamma \min_{i=1,2} Q_{\phi^{(i)}}(\boldsymbol{s}', \boldsymbol{a}'))]^2 \tag{9}$$

$$L(\phi^{(2)}) = \underset{\substack{(\boldsymbol{s}, \boldsymbol{a}, r, \boldsymbol{s}') \sim P_{\mathcal{D}} \\ \boldsymbol{s}' \sim Pr(\cdot | \boldsymbol{s}, \boldsymbol{a}), \\ \boldsymbol{a}' \sim \pi(\cdot | \boldsymbol{s}')}}{\mathrm{E}} [Q_{\phi^{(2)}}(\boldsymbol{s}, \boldsymbol{a}) - (r + \gamma \min_{i=1,2} Q_{\phi^{(i)}}(\boldsymbol{s}', \boldsymbol{a}'))]^2 \tag{10}$$

where $\mathcal{D}$ denotes replay buffer, $P_{\mathcal{D}}$ denotes the distribution that describes the likelihoods of samples drawn from $\mathcal{D}$ uniformly, $Pr(\cdot | \boldsymbol{s}, \boldsymbol{a})$ denotes the transition probability distribution, and $\pi$ denotes the target policy.

### 3.2  AN UPPER BOUND OF THE APPROXIMATION ERROR OF Q FUNCTION

For convenience, we analyze the setting with only one Q function. The concept of domain adaptation is used to describe the task of critic module. The learning task of critic module can be viewed as adapting the learned Q-network to a new Q function for newly learned policy. Thus, naturally, target error, i.e. Eq. (7), equates to the approximation error at current iteration of critic update. In this section, we derive an upper bound of the approximation error of Q function and find that the upper bound of approximation error will be smaller if the more similar the two consecutive policies.

Fig 1 illustrates the training process of an actor-critic method which is a alternating process of value function. At $(n + 1)$-th iteration, the critic module tries to fit the value function, $Q^{\pi_{\theta_n}}$, by means of $\pi_{\theta_n}$. But because of appriximation error, the actually obtained Q-network (approximator), $Q_{\phi_{n+1}}$, is not equal to $Q^{\pi_{\theta_n}}$. This can be expressed by the following equation:

$$Q_{\phi_{n+1}}(\mathbf{s}, \mathbf{a}) = Q^{\pi_{\theta_n}}(\mathbf{s}, \mathbf{a}) + \epsilon_{n+1}^{\mathbf{s}, \mathbf{a}}, \tag{11}$$

where $\epsilon_{n+1}^{\mathbf{s}, \mathbf{a}}$ denotes the appriximation error in Q function given state, $\mathbf{s}$, and action, $\mathbf{a}$. This can be viewed as adapting the learned Q-network, $Q_{\phi_n}$, to the value function, $Q^{\pi_{\theta_n}}$. Hence, the *source* distribution is $P_{\mathcal{D}^n}$, and the *target* distribution is $P_{\mathcal{D}^{n+1}}$. As mentioned at the argument following

Eq. (9) and (10), $P_\mathcal{D}$ denotes the distribution that describes the likelihoods of samples drawn from replay buffer $\mathcal{D}$ uniformly. $\mathcal{D}^n$ and $\mathcal{D}^{n+1}$ are replay buffers at $n$-th and $(n+1)$-th iteration, respectively. $Q_{\phi_n}$ is the *labeling function*. Clearly, *the target error here equals to the approximation error at current iteration of critic update*. This can be expressed by the following equation:

$$e_T(Q_\phi) = \underset{s,a\sim P_{\mathcal{D}^{n+1}}}{E}[|Q_\phi(s,a) - Q^{\pi_{\theta_n}}(s,a)|] = \underset{s,a\sim P_{\mathcal{D}^{n+1}}}{E}[\epsilon_{n+1}^{s,a}]. \tag{12}$$

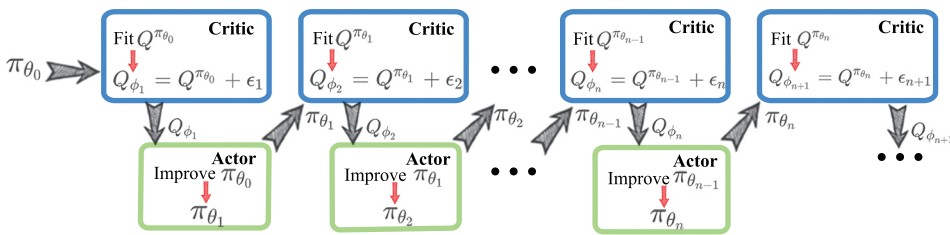

Figure 1: Alternating process of Actor-critic alternates between value function and policy updates.

In addition, because TD method is use to estimate $Q_{\phi_n}$, the real *labeling function* in *target* domain is actually the following one:

$$y_{\pi_{\theta_n}}(s,a) = R_{t+1} + \gamma Q_{\phi_n}(s',a'), s' \sim Pr(\cdot|s,a), a' \sim \pi_{\theta_n}(\cdot|s), \tag{13}$$

where $Pr(\cdot|s,a)$ denotes the transition probability distribution, and $\pi_{\theta_n}$ denotes the policy updated in actor module by using $Q_{\phi_n}$. Hence, the *target error* here means the difference between the Q-network, $Q_\phi(s,a)$, and its target (labeling function) at $(n+1)$-th iteration, $y_{\pi_{\theta_n}}$, on a *target* distribution. The actual target error is given by

$$e_T(Q_\phi) = \underset{s,a\sim P_{\mathcal{D}^{n+1}}}{E}[|Q_\phi(s,a) - y_{\pi_{\theta_n}}(s,a)|] = \underset{\substack{s,a\sim P_{\mathcal{D}^{n+1}}, \\ s'\sim Pr(\cdot|s,a), \\ a'\sim\pi_{\theta_n}}}{E}[|Q_\phi(s,a) - [R_{t+1} + \gamma Q_{\phi_n}(s',a')]|]. \tag{14}$$

Furthermore, two types of error are used to derive upper bound of appximation error, including *source error* $e_S(Q_\phi)$ and error $e_S(Q_\phi, y_{\pi_{\theta_n}})$. *Source error* error here is the difference between the Q-network, $Q_\phi(s,a)$, and its target (labeling function) at the $n$-th iteration, $y_{\pi_{\theta_{n-1}}}$, on a *source* distribution. Hence, the approximation error is given by

$$e_S(Q_\phi) = \underset{s,a\sim P_{\mathcal{D}^n}}{E}[|Q_\phi(s,a) - y_{\pi_{\theta_{n-1}}}(s,a)|] = \underset{\substack{s,a\sim P_{\mathcal{D}^n}, \\ s'\sim Pr(\cdot|s,a), \\ a'\sim\pi_{\theta_{n-1}}}}{E}[|Q_\phi(s,a) - [R_{t+1} + \gamma Q_{\phi_{n-1}}(s',a')]|]. \tag{15}$$

Error $e_S(Q_\phi, y_{\pi_{\theta_n}})$ is the difference between the Q-network, $Q(s,a)$, and its target at $(n+1)$-th iteration, $y_{\pi_{\theta_n}}$, on a *source* distribution, which is given by

$$e_S(Q_\phi, y_{\pi_{\theta_n}}) = \underset{s,a\sim P_{\mathcal{D}^n}}{E}[|Q_\phi(s,a) - y_{\pi_{\theta_n}}(s,a)|] = \underset{\substack{s,a\sim P_{\mathcal{D}^n}, \\ s'\sim Pr(\cdot|s,a), \\ a'\sim\pi_{\theta_n}}}{E}[|Q_\phi(s,a) - [R_{t+1} + \gamma Q_{\phi_n}(s',a')]|]. \tag{16}$$

Target error can be derived into the following inequality:

$$\begin{aligned} e_T(Q_\phi) &= e_T(Q_\phi) + e_S(Q_\phi) - e_S(Q_\phi) + e_S(Q_\phi, y_{\pi_{\theta_n}}) - e_S(Q_\phi, y_{\pi_{\theta_n}}) \\ &\leq e_S(Q_\phi) + |e_S(Q_\phi, y_{\pi_{\theta_n}}) - e_S(Q_\phi)| + |e_T(Q_\phi) - e_S(Q_\phi, y_{\pi_{\theta_n}})| \\ &\leq e_S(Q_\phi) + \underset{s,a\sim P_{\mathcal{D}^n}}{E}[|y_{\pi_{\theta_n}}(s,a) - y_{\pi_{\theta_{n-1}}}(s,a)|] + |e_T(Q_\phi) - e_S(Q_\phi, y_{\pi_{\theta_n}})|, \end{aligned} \tag{17}$$

where the third term in the third line, $|e_T(Q_\phi) - e_S(Q_\phi, y_{\pi_{\theta_n}})|$, is transformed further as:

$$
\begin{aligned}
&|e_T(Q_\phi) - e_S(Q_\phi, y_{\pi_{\theta_n}})| \\
&= \left| \mathop{E}_{\boldsymbol{s},\boldsymbol{a}\sim P_{\mathcal{D}^{n+1}}} [|Q_\phi(\boldsymbol{s},\boldsymbol{a}) - y_{\pi_{\theta_n}}(\boldsymbol{s},\boldsymbol{a})|] - \mathop{E}_{\boldsymbol{s},\boldsymbol{a}\sim P_{\mathcal{D}^n}} [|Q_\phi(\boldsymbol{s},\boldsymbol{a}) - y_{\pi_{\theta_n}}(\boldsymbol{s},\boldsymbol{a})|] \right| \\
&\leq \gamma \mathop{E}_{\boldsymbol{s},\boldsymbol{a}\sim P_{\mathcal{D}^n}} \left| \mathop{E}_{\substack{\boldsymbol{s}'\sim Pr(\cdot|\boldsymbol{s},\boldsymbol{a}),\\ \boldsymbol{a}'\sim\pi_{\theta_n}(\cdot|\boldsymbol{s}')}} [Q_{\phi_n}(\boldsymbol{s}',\boldsymbol{a}')] - \mathop{E}_{\substack{\boldsymbol{s}'\sim Pr(\cdot|\boldsymbol{s},\boldsymbol{a}),\\ \boldsymbol{a}'\sim\pi_{\theta_{n-1}}(\cdot|\boldsymbol{s}')}} [Q_{\phi_n}(\boldsymbol{s}',\boldsymbol{a}')] \right|.
\end{aligned}
\tag{18}
$$

Recall that $\mathcal{D}$ is actually the replay buffer; and $P_{\mathcal{D}}$ denotes the distribution that describes the likelihoods of samples drawn from replay buffer $\mathcal{D}$ uniformly. Because, in experience replay mechanism (Lin, 1992), the number of samples in $\mathcal{D}^{n+1}$ is only a little more than in $\mathcal{D}^n$, the difference between $P_{\mathcal{D}^n}$ and $P_{\mathcal{D}^{n+1}}$ can be ignored. Finally, the upper bound of error in Q-network is determined by

$$
\begin{aligned}
e_T(Q_\phi) \leq\, &e_S(Q_\phi) + \mathop{E}_{\boldsymbol{s},\boldsymbol{a}\sim P_{\mathcal{D}^n}} [|y_{\pi_{\theta_n}}(\boldsymbol{s},\boldsymbol{a}) - y_{\pi_{\theta_{n-1}}}(\boldsymbol{s},\boldsymbol{a})|] \\
&+ \gamma \mathop{E}_{\boldsymbol{s},\boldsymbol{a}\sim P_{\mathcal{D}^n}} \left| \mathop{E}_{\substack{\boldsymbol{s}'\sim Pr(\cdot|\boldsymbol{s},\boldsymbol{a}),\\ \boldsymbol{a}'\sim\pi_{\theta_n}(\cdot|\boldsymbol{s}')}} [Q_{\phi_n}(\boldsymbol{s}',\boldsymbol{a}')] - \mathop{E}_{\substack{\boldsymbol{s}'\sim Pr(\cdot|\boldsymbol{s},\boldsymbol{a}),\\ \boldsymbol{a}'\sim\pi_{\theta_{n-1}}(\cdot|\boldsymbol{s}')}} [Q_{\phi_n}(\boldsymbol{s}',\boldsymbol{a}')] \right|.
\end{aligned}
\tag{19}
$$

It is noticeable that the third term in the upper bound will be smaller if the more similar the two consecutive policies, $\pi_{\theta_{n-1}}$ and $\pi_{\theta_n}$ are. Hence, we can conclude that confining the KL divergence between every two consecutive policies can help limit the approximation error during the optimization process of actor-critic. This conclusion is used to design the learning method of the policy.

## 3.3 ACTOR MODULE–LEARNING A POLICY

The learning task of the actor module is to learn a parameterized stochastic policy, $\pi_\theta(\boldsymbol{a}|\boldsymbol{s})$. In order to lower the upper bound of approximation error of Q function (or to reduce the optimization difficulty of the Q functions), the goal of the policy is converted from maximizing expect discounted return two parts: maximizing the estimated Q values—the minimum of the two Q approximators—and, concurrently, minimizing the KL divergence between two successive policies. The optimization objective is specified by

$$
\max_{\boldsymbol{\theta}} \mathop{E}_{\boldsymbol{s}\sim P_{\mathcal{D}}} [\min_{i=1,2} Q_{\phi_i}(\boldsymbol{s}, \widetilde{\boldsymbol{a}}_{\boldsymbol{\theta}}(\boldsymbol{s})) - D_{KL}(\pi_{\boldsymbol{\theta}}(\cdot|\boldsymbol{s}), \pi_{\boldsymbol{\theta}_{old}}(\cdot|\boldsymbol{s}))],
\tag{20}
$$

where $\mathcal{D}$ is the replay buffer; $P_{\mathcal{D}}$ denotes the distribution that describes the likelihoods of samples drawn from $\mathcal{D}$ uniformly; $\boldsymbol{\theta}$ is the parameters of the policy network; $\boldsymbol{\theta}_{old}$ is the parameters of the policy updated in the last iteration; $\widetilde{\boldsymbol{a}}_{\boldsymbol{\theta}}(\boldsymbol{s})$ is the samples drawn from the target stochastic policy $\pi_{\boldsymbol{\theta}}(\boldsymbol{a}|\boldsymbol{s})$. Note that, in order to back-propagate the error through this sampling operation, we use a Diagonal Gaussian policy and the reparameterization trick, i.e. samples are obtained according to

$$
\widetilde{\boldsymbol{a}}_{\boldsymbol{\theta}}(\boldsymbol{s}) = \mu_{\boldsymbol{\theta}}(\boldsymbol{s}) + \sigma_{\boldsymbol{\theta}}(\boldsymbol{s}) \odot \boldsymbol{\xi}, \quad \boldsymbol{\xi} \sim \mathcal{N}(\boldsymbol{0}, \boldsymbol{I}),
\tag{21}
$$

where $\mu_{\boldsymbol{\theta}}(\boldsymbol{s})$ and $\sigma_{\boldsymbol{\theta}}(\boldsymbol{s})$ are the output of the policy network, and denotes the mean and covariance matrix's diagonal elements, respectively.

The KL divergence between two distributions, for example $p(x)$ and $q(x)$, can be thought of the difference between the cross entropy and entropy, which is specified by

$$
D_{KL}(p||q) = H(p,q) - H(p),
\tag{22}
$$

where $H(p,q)$ denotes the cross entropy between $p(x)$ and $q(x)$, and $H(p)$ denotes the entropy of $p(x)$. In practice, we find it is more effective to minimize the cross entropy between two successive policies and to maximize the entropy of current policy, separately, than to minimize the KL divergence directly. Hence, we expand the original objective Eq. 20 into the following one:

$$
\max_{\boldsymbol{\theta}} \mathop{E}_{\boldsymbol{s}\sim P_{\mathcal{D}}} [\min_{i=1,2} Q_{\phi_i}(\boldsymbol{s}, \widetilde{\boldsymbol{a}}_{\boldsymbol{\theta}}(\boldsymbol{s})) - \alpha H(\pi_{\boldsymbol{\theta}}(\cdot|\boldsymbol{s}), \pi_{\boldsymbol{\theta}_{old}}(\cdot|\boldsymbol{s})) + \beta H(\pi_{\boldsymbol{\theta}}(\cdot|\boldsymbol{s}))],
\tag{23}
$$

where $\alpha$ and $\beta$ denote the coefficients of the cross entropy and the entropy, respectively.

Moreover, we adopt a mechanism to automatically adjust $\alpha$ and $\beta$. To do this, $\alpha$ is adjusted by keeping the current cross entropy close to a target value. This mechanism is specified by the following optimization objective:

$$\min_{\alpha} \mathop{E}_{\boldsymbol{s} \sim P_{\mathcal{D}}} [\log \alpha \cdot ((\delta_{KL} + \delta_{entropy}) - H(\pi_{\boldsymbol{\theta}}(\cdot|\boldsymbol{s}), \pi_{\boldsymbol{\theta}_{old}}(\cdot|\boldsymbol{s})))], \tag{24}$$

where $\delta_{KL}$ denotes the target KL value and $\delta_{entropy}$ denotes the target entropy. Note that $\log(\cdot)$ is use to ensure that $\alpha$ is greater than 0. $\beta$ is adjusted in the same way:

$$\min_{\beta} \mathop{E}_{\boldsymbol{s} \sim P_{\mathcal{D}}} [\log \beta \cdot (H(\pi_{\boldsymbol{\theta}}(\cdot|\boldsymbol{s})) - \delta_{entropy})]. \tag{25}$$

The overall training process is summarized in Appendix A, and code can be found on our GitHub https://github.com/SingerGao/ECAC.

## 4 EXPERIMENTS

The experiments aim to evaluate the effectiveness of the proposed strategy to lower the approximation error and to verify that the proposed method can outperform other mainstream RL algorithms. The experiments for ablation study and comparative evaluation are conducted on a few challenging continuous control tasks from the OpenAI Gym(Brockman et al., 2016) environments, which includes Mujoco(Todorov et al., 2012) and Pybullet (Coumans & Bai, 2016–2019) versions. Implementation details and hyperparameters of ECAC are presented in Appendix B.

### 4.1 ABLATION STUDY

Ablation studies are performed to verify the contribution of the operation of KL limitation. We compared the performance of ECAC with the method removing KL limitation from ECAC.

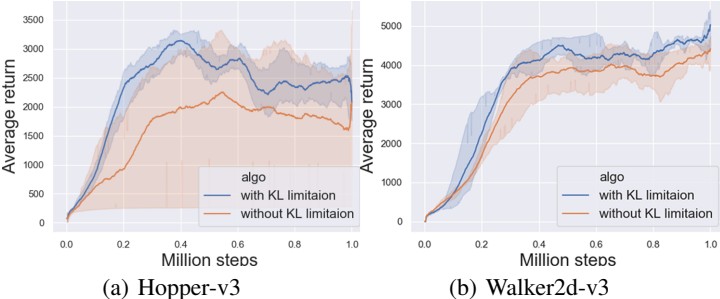

(a) Hopper-v3        (b) Walker2d-v3

Figure 2: Performance comparison of the method with KL limitation and the one without KL limitation on the Hopper-v3 and Walker2d-v3 benchmark. The method with KL limitation performs better. Curves are smoothed uniformly for visual clarity.

Figure 2 compares five different instances for both methods with and without KL limition using different random seeds; and each instance performs five evaluation episodes every $1,000$ environment steps. The solid curves corresponds to the mean and the shaded region to the minimum and maximum returns over the five runs. The experimental result shows that our method with KL limitation performs better than the one without KL limitation. Figure 3 demonstrates that by using ECAC the KL divergence remains comfortably low during all the training process.

Furthermore, to verify that confining the KL divergence can decrease the approximation error in Q function, we measured the normalized approximation error in 100 random states every $10,000$ environment steps. The normalized approximation error is specified by

$$e_Q = \frac{|Q_{approx} - Q_{true}|}{|Q_{ture}|} \tag{26}$$

where $Q_{approx}$ is the approximate Q value given by the current Q network, and $Q_{ture}$ is the true discounted return. The true value is estimated using the average discounted return over 100 episodes

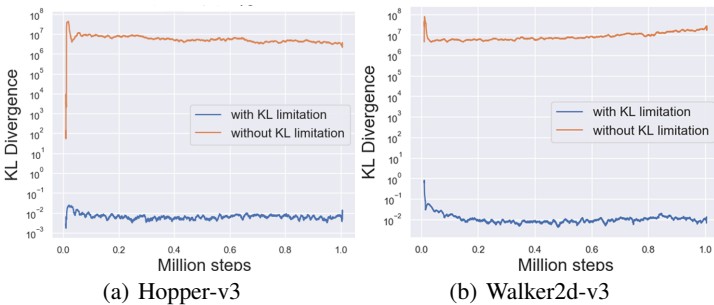

(a) Hopper-v3      (b) Walker2d-v3

Figure 3: The KL divergence comparison of the method with KL limitation and the one without KL limitation on the Hopper-v3 and Walker2d-v3 benchmark.

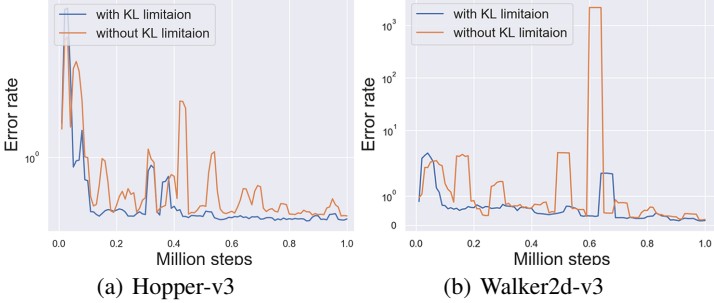

(a) Hopper-v3      (b) Walker2d-v3

Figure 4: The approximation error comparison of the method with KL limitation and the one without KL limitation on the Hopper-v3 and Walker2d-v3 benchmark.

following the current policy, starting from states sampled from the replay buffer. Figure 4 shows that the method with KL limitation has lower error in the Q function.

The results of all the ablation studies indicates that the approximation error of Q functioncan be decreased and the performance of the RL algorithm can be improved by placing restrictions on KL divergence between every two consecutive policies.

## 4.2 COMPARATIVE EVALUATION

Comparative evaluation are conducted to verify that our method can outperform other traditional RL methods including A2C, PPO, TD3, and SAC. Five individual runs of each algorithm with different random seeds are done; and each run performs five evaluation episodes every $1,000$ environment steps. Our results are reported five random seeds (one random seed for one individual run) of the Gym simulator, the network initialization, and sampling actions from policy during the training.

The results of max average return over five runs on all the 10 tasks are presented in Table 1. ECAC outperforms all other algorithms on the tasks except Hopper-v3 and HumanoidBulletEnv-v0 are only next to TD3 on Hopper-v3 and HumanoidBulletEnv-v0. Figure 5 and Figure 6 demonstrates learning curves of comparative evaluation on the 10 continuous control tasks (Mujoco and PyBullet version, respectively).

## 5 CONCLUSION

This paper presented a model-free actor-critic method based on a finding that the approximation error in value function of RL methods can be decreased by placing restrictions on KL-divergence between every two consecutive policies. Our method increases the similarity between every two consecutive polices in the training process and therefore reduces the optimization difficulty of value function. In the ablation studies, we compare the approximation error in Q function, KL divergence, and performance of the methods with and without KL limitation. The results of ablation study show that the proposed method can decrease the approximation error and improved the performance. Moreover, the results of comparative evaluation demonstrate that ECAC outperforms other model-free deep RL algorithm including A2C, PPO, TD3, and SAC.

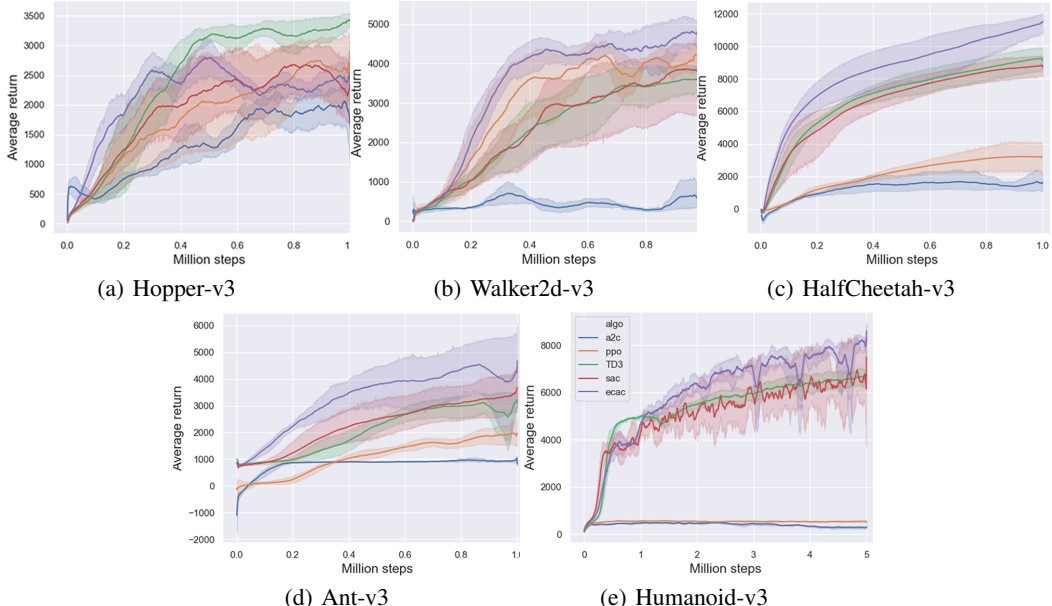

Figure 5: The results of comparative evaluation on Mujoco version of the OpenAI gym continuous control tasks. Curves are smoothed uniformly for visual clarity.

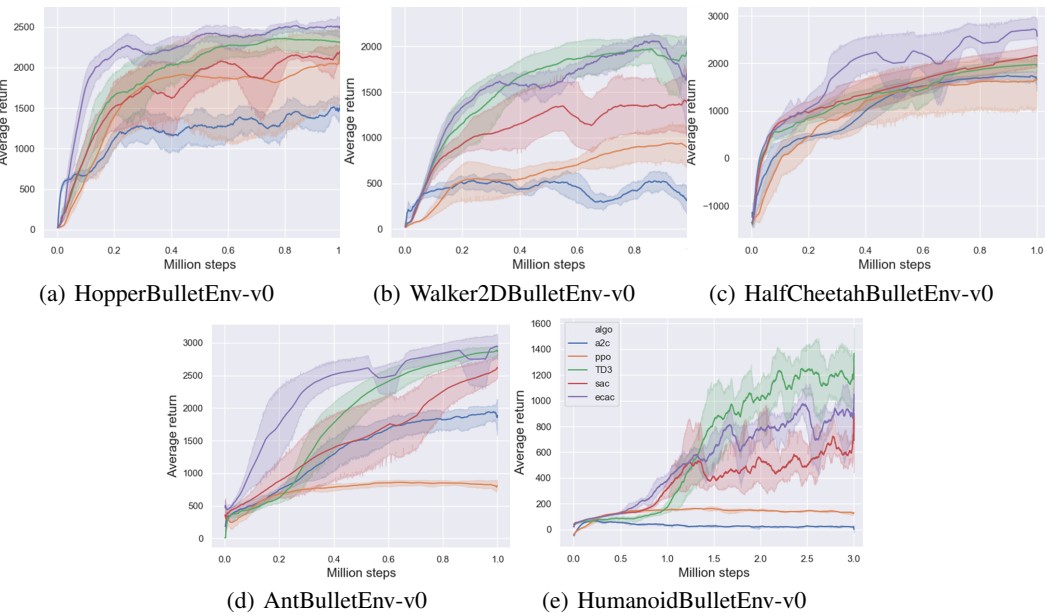

Figure 6: The results of comparative evaluation on Pybullet version of the OpenAI gym continuous control tasks. Curves are smoothed uniformly for visual clarity.

Table 1: Max average return over five runs on all the 10 tasks. Maximum value for each task is bolded. $\pm$ corresponds to standard deviation over runs.

| Environment (Total steps) | ECAC | A2C | PPO | TD3 | SAC |
|---|---|---|---|---|---|
| Hopper ($10^6$) | $3395.4 \pm 69.5$ | $2875.8 \pm 118.0$ | $3317.6 \pm 150.3$ | $\mathbf{3493.6 \pm 134.2}$ | $3175.2 \pm 110.0$ |
| Walker2d ($10^6$) | $\mathbf{5146.2 \pm 243.8}$ | $2479.4 \pm 561.7$ | $4811.1 \pm 295.7$ | $3960.3 \pm 382.5$ | $4455.1 \pm 945.4$ |
| HalfCheetah ($10^6$) | $\mathbf{11744.7 \pm 746.9}$ | $2653.5 \pm 975.9$ | $3422.1 \pm 1039.4$ | $9492.6 \pm 852$ | $9211.6 \pm 1051.3$ |
| Ant ($10^6$) | $\mathbf{5420.8 \pm 1069.6}$ | $1651.6 \pm 140.5$ | $2502.1 \pm 442.8$ | $3462.4 \pm 488.5$ | $3832.2 \pm 927.7$ |
| Humanoid ($5 \cdot 10^6$) | $\mathbf{8953.4 \pm 244.8}$ | $751 \pm 34.2$ | $844.2 \pm 83.2$ | $6969.2 \pm 403.2$ | $8265.8 \pm 937.1$ |
| HopperBulletEnv ($10^6$) | $\mathbf{2642.5 \pm 39.3}$ | $1688.7 \pm 68.9$ | $2255.5 \pm 342.9$ | $2488.4 \pm 172.6$ | $2397.7 \pm 56.9$ |
| Walker2DBulletEnv ($10^6$) | $\mathbf{2323 \pm 161.8}$ | $1005.4 \pm 5.7$ | $1035 \pm 201.2$ | $2116.6 \pm 151.9$ | $1651.6 \pm 408.9$ |
| HalfCheetahBulletEnv ($10^6$) | $\mathbf{2794.9 \pm 281.8}$ | $2027.7 \pm 87.8$ | $1730 \pm 639$ | $2012.8 \pm 182.2$ | $2202.4 \pm 243$ |
| AntBulletEnv-v0 ($10^6$) | $\mathbf{2997.6 \pm 220.9}$ | $2292.4 \pm 175.1$ | $969.7 \pm 31.5$ | $2953.9 \pm 84.3$ | $2650.2 \pm 188$ |
| HumanoidBulletEnv ($3 \cdot 10^6$) | $1226.7 \pm 45.3$ | $110.5 \pm 5.4$ | $208.6 \pm 10.9$ | $\mathbf{1471.1 \pm 113.9}$ | $1052.5 \pm 85.6$ |

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

APPENDICES

## A  PSEUDO CODE OF ECAC

---

**Algorithm 1** Error controlled Actor-critic.

---

**Require:** initial policy patameters, $\boldsymbol{\theta}$; Q function parameters, $\phi_1$ and $\phi_2$; discount rate, $\gamma$; the coefficient of KL term, $\beta$; empty replay buffer $\mathcal{D}$; the number of episodes, M; the maximum number of steps in each episode, T; minibatch size, N.

**Ensure:** optimal policy parameters $\boldsymbol{\theta}^*$.

1: **for** $episode = 1, \text{M}$ **do**
2:    Reset environment.
3:    **for** $t = 1, \text{T}$ **do**
4:        Obeserve state $\boldsymbol{s}$ and select action $\boldsymbol{a} \sim \pi(\cdot|\boldsymbol{s})$.
5:        Execute $\boldsymbol{a}$ in the enviroment.
6:        Observe next state $\boldsymbol{s}'$ and reward $r$.
7:        Store transition $(\boldsymbol{s}, \boldsymbol{a}, r, \boldsymbol{s}')$ in replay buffer $\mathcal{D}$.
8:        Randomly sample a minibatch of N transitions, $B = \{(\boldsymbol{s}, \boldsymbol{a}, r, \boldsymbol{s}')\}$ from $\mathcal{D}$.
9:        Compute targets for the Q functions:

$$y(r, \boldsymbol{s}') = r + \gamma \min_{i=1,2} Q_{\phi_i}(\boldsymbol{s}', \widetilde{\boldsymbol{a}}'), \ \widetilde{\boldsymbol{a}}' \sim \pi_\theta(\cdot|\boldsymbol{s}').$$

10:        Update Q functions by one step of gradient descent using

$$\nabla_\phi \frac{1}{N} \sum_{(\boldsymbol{s}, \boldsymbol{a}, r, \boldsymbol{s}') \in B} (Q_{\phi_i}(\boldsymbol{s}, \boldsymbol{a}) - y(r, \boldsymbol{s}'))^2, \ i = 1, 2.$$

11:        Backup old policy, $\boldsymbol{\theta}_{old} \leftarrow \boldsymbol{\theta}$.
12:        Update $\alpha$ and $\beta$ by one step of gradient ascent using

$$\nabla_\alpha \frac{1}{N} \sum_{s \in B} [\log \alpha \cdot ((\delta_{KL} + \delta_{entropy}) - H(\pi_{\boldsymbol{\theta}}(\cdot|\boldsymbol{s}), \pi_{\boldsymbol{\theta}_{old}}(\cdot|\boldsymbol{s})))],$$

$$\nabla_\beta \frac{1}{N} \sum_{s \in B} [\log \beta \cdot (H(\pi_{\boldsymbol{\theta}}(\cdot|\boldsymbol{s})) - \delta_{entropy})],$$

where $\delta_{KL}$ and $\delta_{entropy}$ denote target KL divergence and target entropy, respectively.
13:        Update policy by one step of gradient ascent using

$$\nabla_{\boldsymbol{\theta}} \frac{1}{N} \sum_{s \in B} (Q_{\phi_1}(\boldsymbol{s}, \widetilde{\boldsymbol{a}}_{\boldsymbol{\theta}}(\boldsymbol{s})) + \alpha H(\pi_{\boldsymbol{\theta}}(\cdot|\boldsymbol{s})) - \beta D_{KL}(\pi_{\boldsymbol{\theta}}(\cdot|\boldsymbol{s}), \pi_{\boldsymbol{\theta}_{old}}(\cdot|\boldsymbol{s}))),$$

where $\widetilde{\boldsymbol{a}}_{\boldsymbol{\theta}}(s)$ is a sample from $\pi_{\boldsymbol{\theta}}(\cdot|\boldsymbol{s})$, which is differentiable with respect to $\theta$ via the reparameterization trick.
14:    **end for**
15: **end for**

---

Table 2: ECAC Hyperparameters

| Parameter | Value |
|---|---|
| learning rate | $10^{-3}$ |
| discount($\gamma$) | 0.99 |
| replay buffer size | $5 \cdot 10^5$ |
| batch size | 128 |
| target KL | $5 \cdot 10^{-3}$ |
| target entropy | $-dim(\mathcal{A})/2.0$ (e.g. , $-3$ for Walker2d-v3) |
| target smoothing coefficient ($\tau$) | $5 \cdot 10^{-3}$ |
| number of hidden layers (all networks) | 2 |
| number of hidden units per layer | 256 |
| nonlinearity | ReLU |

Table 3: Reward Scale Parameter

| Environment | Reward Scale |
|---|---|
| Hopper-v3 | 5 |
| Walker2d-v3 | 5 |
| HalfCheetah-v3 | 5 |
| Ant-v3 | 5 |
| Humanoid-v3 | 20 |
| HopperBulletEnv-v0 | 5 |
| Walker2DBulletEnv-v0 | 5 |
| HalfCheetahBulletEnv-v0 | 5 |
| AntBulletEnv-v0 | 5 |
| HumanoidBulletEnv-v0 | 20 |

# B IMPLEMENTATION DETAILS AND HYPERPARAMETERS OF ECAC

For the implementation of ECAC, a two layer feedforward neural network of 256 hidden units, with rectified linear units (ReLU) between each layer are used to build the two Q functions an the policy. The parameters of the neural networks and the two coefficients (i.e $\alpha$ and $\beta$) are optimized by using Adam(Kingma & Ba, 2014). The hyperparameters of ECAC are listed in Table 2. Moreover, we adopt the target network technique in ECAC, which is common in previous works (Lillicrap et al., 2016; Fujimoto et al., 2018). We also adopt reward scale trick which is presented in Haarnoja et al. (2018); and the reward scale parameter is listed in Table 3.

