# OpenReview forum: "Error Controlled Actor-Critic Method to Reinforcement Learning"
_ICLR.cc/2021/Conference — Reject_

### Official Review · AnonReviewer3 · 2020-10-24
**Good paper starting from theoretical analysis to practical algorithms**

**Rating:** 5
**Confidence:** 4

**Review:**

In this paper, the authors study the error introduced by the estimation of critic function in the Actor-Critic algorithm. Then the author proposed an algorithm that utilizes the idea of double Q learning and using a KL-divergence like regularization method to control this error. Experimentally the proposed algorithm achieves good results comparing to the vanilla Actor-Critic algorithm. This paper shows a successful routine from the theoretical analysis to a practical algorithm.

However, one of the drawbacks of this paper to me is, the authors proposed the double Q learning method. It is not clear how this double Q help to reduce the estimation error? I am also a little bit concerned that the gap between theoretical analysis and the practical algorithm is large. In practice, the authors always use a one-step update, but there is no guarantee that the one-step update for both actor and critic can decrease the error or improve the performance.

Also, since the key idea of how to control the critic is to use a regularization based on the last policy to make sure the policy move not so 'fast'. It would be interesting to ask what will happen if we only use a smaller learning rate for the actor and a larger learning rate for the critic. Some theoretical work [1] [2] have already adopted this method to get the error controlled and make the actor converge to the stationary point. Therefore, it would highlight the result of this paper if the author can compare their algorithm and show the advantage over these algorithms that only set different learning rates.

Given the contribution of the newly proposed algorithm, I will suggest accepting this paper.

[1] Wu, Yue, et al. "A Finite Time Analysis of Two Time-Scale Actor Critic Methods." arXiv preprint arXiv:2005.01350 (2020).

[2] Hong, Mingyi, et al. "A two-timescale framework for bilevel optimization: Complexity analysis and application to actor-critic." arXiv preprint arXiv:2007.05170 (2020).

--- Update after discussion

I agree with other reviewers' idea that the contribution of this paper is somehow limited. Since my previous suggestion (7, accept) is based on that the author can successfully address how to control the error of the critic theoretically. However, with only the experimental elaboration study, as I said in the response to the author, the contribution of this paper is limited. Thus, I would switch back to 5 (marginally reject) based on that after reading other reviewers' discussion.

---

> ### Author Response · Authors · 2020-11-23
> **Responses to the comments**
>
> ### Comment #1:
>
> However, one of the drawbacks of this paper to me is, the authors proposed the double Q learning method. It is not clear how this double Q help to reduce the estimation error?
>
> ### Response #1:
>
> As mentioned at the beginning of Sec 4.1 in the initial submission, (sec 3.1 in the revised paper), we adopt the clipped double-Q strategy to reduce overestimation, which was proposed by Fujimoto [1]. The key idea of it is to decrease the likelihood of overestimation by increasing the likelihood of underestimation.
>
> The main idea of our method is to derive an upper boundary of approximation errors for Q function approximator in actor-critic methods, and to reduce the approximation error by minimizing this upper bound of the error.  This is similar to the Expectation-Maximization Algorithm (EM) [2] which maximizes a lower bound of the log-likelihood instead of the log-likelihood directly.
>
> Thank you for this comment, in the revision, we have added a very brief discussion about how clipped double Q reduce overestimation.
>
> ### Comment #2:
>
> I am also a little bit concerned that the gap between theoretical analysis and the practical algorithm is large. In practice, the authors always use a one-step update, but there is no guarantee that the one-step update for both actor and critic can decrease the error or improve the performance.
>
> ### Response #2:
>
> The main idea of our method is to reduce the approximation error by minimizing a derived upper bound of the error (similar to the EM algorithm). We find that the KL penalty is an effective way. Even using a one-step update, our method indirectly stabilizes the Q function by using the KL penalty when training the policy during the whole training process. The presented results of our ablation study showed that the KL penalty reduced the approximation error (which is shown in Fig 4 in the revision).
>
> ### Comment #3:
>
> Also, since the key idea of how to control the critic is to use a regularization based on the last policy to make sure the policy move not so 'fast'. It would be interesting to ask what will happen if we only use a smaller learning rate for the actor and a larger learning rate for the critic. Some theoretical work [1] [2] have already adopted this method to get the error controlled and make the actor converge to the stationary point.
>
> ### Response #3:
>
> Thank you very much for recommending two good papers on the two-time-scale actor-critic method. The idea of two-time-scale methods seems similar to the delayed policy update method [1] which only updates the policy after a fixed number of updates to the Q network. Delayed policy update is effective, but the benefits are limited. We will compare these works to KL-penalty in the future work. Thank you!
>
> ## Reference
>
> [1] Scott Fujimoto, Herke van Hoof, and David Meger. Addressing function approximation error in actor-critic methods. ICML 2018.
>
> [2] Christopher M Bishop. *Pattern Recognition and Machine Learning (Information Science and Statistics)*. Springer-Verlag New York, Inc., 2006.

---

> > ### Comment · AnonReviewer3 · 2020-11-25
> > **Thanks for your response**
> >
> > Thanks for your response. In my previous review, my major concern is the gap between the theory and practice, like how the error is controlled and how can the one-step optimization be guaranteed to reduce the error. The authors' response tried to do some ablation study on that. However, I'm desiring more theoretical solutions or guarantees for that. Without that, the author's contribution is somehow limited.

---

> > > ### Author Response · Authors · 2020-11-25
> > > **Thank you for your reply!**
> > >
> > > Thank you for your suggestions!

---

### Official Review · AnonReviewer4 · 2020-10-26
**Seems to be a heuristic (perhaps not entirely new) presented in a not so clear manner**

**Rating:** 3
**Confidence:** 5

**Review:**


Clarity:
---------
Major:
******
It is not clear what the actor critic algorithm is. What is presented in Fig 1 seems to be policy iteration, and Theorem 1 seems to be stating policy iteration (yet $\pi^{n+1}$ has not been defined).

Minor:
*****
1) What is $V^{\pi^n}$? Is it the value function of the policy $\pi^n$.
2) What is the difference between $\pi^{*,n}$ and $\pi^{n+1}$.
3) What is $\pi$ is eq(3)?


Quality:
-----------
1) The argument following eq (15) is very informal: "the number of samples in $\mathcal{D}^{n+1}$ is only a little more than in $\mathcal{D}^n$".
2) For the contraction property of the Bellman operator, the author can consider citing standard text books instead of "https://towardsdatascience.com/ mathematical-analysis-of-reinforcement-learning-bellman-equation-ac9f0954e19f"


Significance:
------------------
1) The  authors claim "We present a convergence analysis for actor-critic methods by Banach’s Fixed Point Theorem" to be one of the contributions. The fixed point theorem here is to show that Bellman operator is a contraction operator (a well known fact).
2) The experiments are on only 5 domains.


Overall Feedback: Adding the KL based penalty is perhaps a useful idea (this may also not be an entirely new one). It will be great if a) the algorithm can clearly stated,  b) the novelty of the method pointed out clearly by comparing it with related works c) the ideas can be supported by formal theoretical statements.

---

> ### Author Response · Authors · 2020-11-21
> **Thank you for your comments, explain some unreasonable comments, and have improved clarity in the revised paper.**
>
> Sorry for my writing mistakes. We organized all the comments into the following three categories:
> ## Reasonable
> ### Comment #1:
> The argument following eq (15) is very informal...
> ### Response #1:
> We have revised the equation (Eq 18 in the revised paper) and the argument by following this comment.
> ### Comment #2 :
> (1) What is $V^{π^{n}}$? Is it the value function of the policy $\pi^n$.
> (2) What is the difference between $π^{∗,n}$ and $π^{n+1}$.
> (3) What is π is eq(3)?
> ### Response #2:
> We have improved clarity in the revised paper.  (Secs 2 and 3.2 in the revised paper)
>
>
> ## Unreasonable
> ### Comment #3:
> It is not clear what the actor critic algorithm is. What is presented in Fig 1 seems to be policy iteration, and Theorem 1 seems to be stating policy iteration (yet πn+1 has not been defined).
> ### Response #3:
> Actor-critic ([3], [4], and [5]) is a well-known architecture in RL. Due to the paper limitation, we only provided a very brief introduction in our initial submission. We have provided a more clear introduction to actor-critic in the revised paper (sec 2.2).
>
> - What is presented in Fig 1 is certainly actor-critic. An actor-critic method ([3], [4], and [5])  is an alternating update algorithm. Besides, we have updated Fig. 1 to make it more clear.
>
> - Theorem 1 is surely stating actor-critic.  Please do carefully read our convergence proof (Appendix A in the first submission) .
>
> ### Comment #4:
> Adding the KL based penalty is perhaps a useful idea
> ### Response #4:
> We have updated the paper by adding KL-penalty literatures, [1] and [2], in the revised paper (sec 1). These works are different from ours. The **difference between the KL-control literatures and our work** are summarized as follows:
>
> - We arrived at the same conclusion as [1] and [2] by a somewhat **different route**.  The authors **directly studied the effect of KL and entropy regularization in RL** and proved that a KL regularization indeed leads to averaging errors made at each iteration of value function update.  While our idea is very different from theirs: It is impracticable to minimize the approximation error directly, so  instead we tried to **minimize an upper bound of approximation error**.
>
> - We arrived at **a more general conclusion**: approximation error can be reduced by keep new policy close to previous one.  Please see page 4 in the revised paper, the third term in the upper bound of approximation error will be smaller if the more similar the two consecutive policies,  $\pi_{\theta_{n}}$ and $\pi_{\theta_{n-1}}$, are.  **KL penalty is a effective way, but not the only way.**
>
> - [2] used KL penalty to improve the performance of DQN, while we proposed a more practical method (i.e. ECAC) which includes  a mechanism to automatically adjust the coefficient of KL term.
>
>
> ### Comment #5 :
> The authors claim "We present a convergence analysis for actor-critic methods by Banach’s Fixed Point Theorem" to be one of the contributions. The fixed point theorem here is to show that Bellman operator is a contraction operator (a well known fact).
> ### Response #5:
> The reviewer fully misunderstood our idea. Our convergence proof (see Appendix A in the first submission) consists of two steps:
>
> (1) Prove the actor-critic operator is a "contraction", which is one of the sufficient conditions of Fixed Point Theorem. **Notice that the actor-critic operator, which includes an operation to improve the policy (i.e. Eq. 3), is not equal to the Bellman operator!**
>
> (2) Prove actor-critic methods fit Banach's Fixed Point Theorem, thus actor-critic methods converge to a unique optimal policy.
> Besides, **we have removed the convergence analysis** (Sec 3 in the first submission), this does not affect the main idea of our work (i.e. minimize an upper bound of approximation error).
>
> ### Comment #6 :
> The experiments are on only 5 domains.
> ### Response #6:
> - We demonstrated the experimental results on **TEN** control tasks (**five of them are in Appendix D in the initial submission**).  For a conference paper, it is common to conduct experiments on only the most difficult tasks. (see TD3 [7] and SAC [8]). In the revised paper, we have moved the five results in the main body of the paper.
>
> ## Reference
> [1] Matthieu Geist  et al. A theory of regularized markov decision processes, 2019.
>
> [2] Nino Vieillard  et al. Leverage the average: an analysis of kl regularization in rl, 2020.
>
> [3] Richard S. Sutton and Andrew G. Barto. Reinforcement Learning: An Introduction (Second Edition). 2018.
>
> [4] Konda, V.R. and Tsitsiklis, J.N. Actor-Critic Algorithms. NIPS.1999.
>
> [5] Thomas Degris  et al. Off-policy actor-critic, 2013.
>
> [6] Agarwal, Praveen et al. "Banach Contraction Principle and Applications". Fixed Point Theory in Metric Spaces.
>
> [7] Scott Fujimoto et al. Addressing function approximation error in actor-critic methods. ICML 2018.
>
> [8] Tuomas Haarnoja et al. Soft actor-critic. ICML 2018.

---

> > ### Comment · Area_Chair1 · 2020-11-24
> > **Please engage in the conversation**
> >
> > Dear reviewer,
> >
> > Please let us know whether the authors' rebuttal to your questions are satisfactory or not. If you need more clarifications on any issue, please ask your questions as soon as possible. Today (Nov. 24) is the last day that the authors can reply back to you.
> >
> > Thank you,
> > AC

---

> > ### Comment · AnonReviewer4 · 2020-11-24
> > **Thanks for the response.**
> >
> > I hereby acknowledge that I have read the author response.

---

> > > ### Author Response · Authors · 2020-11-24
> > > **Thank you for your reply! We are looking for your feedback!**
> > >
> > > Please let us know whether our rebuttal to your questions are satisfactory or not.
> > > Thank you very much!

---

### Official Review · AnonReviewer1 · 2020-10-30
**Lack of novelty, Inadequate comparison to prior work, Suboptimal baselines**

**Rating:** 3
**Confidence:** 5

**Review:**

Before beginning with the review, I would strongly request the authors to correct grammar errors, spelling errors, and notation inconsistencies which seriously hinder understanding in some cases.

This paper explores error accumulation in actor-critic methods. The authors claim to present an analysis of approximation error in actor-critic methods (though I couldn't find any concrete analysis of this fact) and then suggest that bounding the KL-divergence of the policy against the previous policy aids actor-critic algorithms by deriving an upper bound on the change in TD error due to faster changes in the policy.

My overall opinion is that this paper restates results that are already well known and the general theme of the method ECAC is already explored in RL literature (see KL-control literature). So, I don't really think there is any novel concept presented in the paper. I move to specific points next:

- Theorem 1: I don't see what's novel here. Such results have been shown in the absence of function approximation for off-policy actor-critic (Degris et al. 2012). What's new here?

- Analysis of approximation error: In the abstract, it is claimed that the paper performs an *analysis* of this error. I could not find this anywhere in this paper. If this refers to Section 3.2 (Hidden dangers in AC methods), I would not call that "analysis". It states some known facts without actually discussing anything formally there. So, either this claim should be removed or a valid analysis needs to be presented.

- The reason for why KL-constraints work: I can see what the derivation does and how the authors arrive at the conclusion of KL-constraint I do not buy the argument that KL-constraints help learning because they control TD error -- they do stabilize the critic, as would a smaller learning rate for the policy would and lead to lower accumulation of error (for e.g., see A theory of Regularized MDPs, Geist et al. or Leverage The Average: An Analysis of KL regularization in RL) and these analyses are far more sophisticated that the argument presented in this paper.

- The empirical results in the paper are OK, but the baselines utilized are a bit suboptimal, for e.g. SAC and TD3 in HalfCheetah -- having used these myself, it seems like official codebases of these papers do get better performance than what is reported.  I think the numbers are this unreliable.

I think overall the algorithm with the KL-constraint is a good idea -- as has been explored several times in literature -- but I don't think this paper cites that work, nor compares to them along with not a rigorous enough evaluation and poor writing quality.

---

> ### Author Response · Authors · 2020-11-18
> **Thank the reviewer for the review and expect a more reasonable review!**
>
> We organized all comments as follows:
>
>
>
> **comment#1**:KL-constraint is a good idea--as has been explored several times in literature -- but I don't think this paper cites that work...
>
> **Response#1**: Sorry for this mistake.  We have discussed the suggested papers  (Geist et al. 2019 and Vieillard et al. 2020)  in the revision.  The **difference between the KL-control papers and our work** are summarized as follows:
>
> - Our paper conducted the same conclusion as the KL-control papers by a **different route**.  The authors **directly studied the effect of KL and entropy regularization in RL** and proved that a KL regularization indeed leads to averaging errors made at each update iteration of the value function. While our idea is very different from theirs': It is impracticable to minimize the approximation error directly, instead we tried to **minimize an upper bound of approximation error** (this is similar to the Expectation-Maximization Algorithm (EM) which maximizes a lower bound of the log-likelihood instead of the log-likelihood directly). We derived **an upper boundary of approximation errors for Q function approximator** in actor-critic methods, and found that the error can be reduced by retaining a new policy close to the previous one.
>
> - Our work produced **a more general conclusion**: approximation errors can be reduced by retaining a new policy close to the previous one.  The third term of Eq.19 in the upper bound of approximation errors will be smaller if the more similar the two consecutive policies, $\pi_{\theta_{n}}$ and $\pi_{\theta_{n-1}}$ are.  **The KL penalty is not the only way**.
>
> - Vieillard et al. used the KL penalty to improve the performance of a DQN, while we proposed a better method (i.e. ECAC) which includes a mechanism to *automatically adjust the coefficient of KL term*.
>
> **comment#2**: In the abstract, it is claimed that the paper performs an *analysis* of this error. I could not find this anywhere in this paper. If this refers to Section 3.2 (Hidden dangers in AC methods), I would not call that "analysis".
>
>   **response#2**: Based on this comment, we decide to delete Section 3 because this does not affect main idea of our work, i.e. minimize an upper bound of approximation errors.
>
>
> **comment#3**:Theorem 1: .... Such results have been shown in the absence of function approximation for off-policy actor-critic (Degris et al. 2012). What's new here?
>
> **response#3**:Degris et al. presented convergence proofs of off-policy actor-critic **with a linear approximation of value function**. In contrast, we provide a convergence proof for actor-critic methods using function approximation (**might be linear or non-linear approximation**).  However, for the clarity of this paper, we deleted Section 3.
>
> ## Unreasonable comments:
>
> **comment#4**: The reason for why KL-constraints work: I can see what the derivation does and how the authors arrive at the conclusion of KL-constraint I do not buy the argument that KL-constraints help learning because they control TD error
>
> **response#4**: In the first submission, we did not claim that "KL-constraints help learning because they control TD error"; however, we do arrive at the conclusion that approximation error (**rather than TD error !**) can be reduced by retaining a new policy close to the previous one.  Therefore, our main idea is to **minimize an upper bound of approximation error to reduce approximation error**.
>
> **comment#5**:but the baselines utilized are a bit suboptimal, .... .I think the numbers are this unreliable.
>
> **response#5**:  This comment is **an absolutely unreasonable comment**! It seems the reviewer did not have too much experience with RL experiments.
>
> - The performance of RL algorithms can be drastically different by **using different random seed**. Please refer *Deep Reinforcement Learning that Matters* Henderson et al. 2017. While, to ensure fairness, it is a common way to run five experimental trials for each evaluation, each trial with a different preset random seed (**all algorithms in comparative evaluation used the same set of random seeds**). **That is why the baselines are slightly different from SAC and TD3 papers**.
> - Even if we compare the results presented in our paper with the results reported in other SAC and TD3 papers (an unfair way due to different random seeds), **ECAC still outperforms other algorithms**. For example,  SAC scored nearly 11,000 while ECAC scored 12,000 in HalfCheetah at around 1 millionth step.
> - We hope the reviewer will run an check our code [https://github.com/SingerGao/ECAC ] (**no any author information**) and take back this unreasonable comment.
>
> ## Summary
> **We thank the reviewer for the comments and expect a more reasonable review**. Sorry for my poor English writing skill. I'm not a native English speaker.  I do expect to participate in international academic communications by writing English papers.

---

> > ### Comment · Area_Chair1 · 2020-11-24
> > **Please engage in the conversation**
> >
> > Please let us know whether the authors' rebuttal to your questions are satisfactory or not. If you need more clarifications on any issue, please ask your questions as soon as possible. Today (Nov. 24) is the last day that the authors can reply back to you.
> >
> > Thank you,
> > AC

---

> > ### Comment · AnonReviewer1 · 2020-11-24
> > **Thank you for the response**
> >
> > Thank you for the response. I still have some concerns as follows:
> >
> > 1. It is good that that authors now compare with the papers I mentioned. However, now I am wondering as to what are the consequences of minimizing approximation error? Does it give an upper bound on total error accumulated? Conventional analysis of dynamic programming capture this quantity and it is hard for me to relate the approximation error discussed here to any such analysis. If the authors can elaborate on this point, I can perhaps understand better, but right now it comes across as disconnected from other works.
> >
> > 2. Can the authors elaborate on the point about connection to EM? Is this a formal connection?
> >
> > 3. I don't think that those prior works rely heavily on the KL-penalty -- any penalty that is similar to the KL should also work, so I am not sure about that part in the rebuttal.
> >
> > 4. The auto-tuning the penalty was also done in SAC in the same way, so I am not sure why this is novel. Perhaps the authors can explain.
> >
> > 5. It seems like Theorem 1 and Section 3 were removed (they are not there in the paper) anymore.
> >
> > 6. Having used a number of baselines myself, and from multiple repositories, it does seem to me that using the right set of baselines is needed -- often a times atleast for me and researchers I know, varying trends can be obtained if the choice of baseline algorithms is not the most optimal. So, I am sorry if the point seemed unreasonable, but I am not sure I see a huge benefit on already published algorithms (for example, if I compare to the SAC paper).
> >
> > My current thoughts are that that even if I discount the point 6 above and even if I discount writing concerns (It did not affect my review in any way above, I wrote it as suggestion for improvement), the idea of KL constraint is not new (and the paper still doesn't cite several prior papers on KL-control, e.g., Jaques et al. 2019 (Way Off-Policy...),  G-Learning, etc), and the theoretical motivation behind this idea presented in the paper doesn't seem strong to me. As I said in my review, there should also have been a theoretical/empirical discussion/comparison of alternate techniques like slower learning rate for the policy rather than an explicit KL constraint.
> >
> > To summarize, I still don't see why the perspective on this paper is useful beyond already exisitng perspectives and what it implies for a holistic bound (e.g., for compounding errors). Since that's the only novel part of the paper and as KL-control and the KL-constrained actor-critic algorithm has been known for a long time, I am inclined to base my score on the theoretical part. The paper doesn't discuss any new design choices/ algorithmic decisions (the auto-tuning KL penalty was done in SAC too, so it doesn't seem like a contribution by this paper).  Hence, I retain my score for now. I am happy to reconsider if the authors feel some of these points are incorrect/unreasonable.

---

> > > ### Author Response · Authors · 2020-11-24
> > > **Responses to new comments, we're looking forward to your reply!**
> > >
> > > Thank you for your reply!!
> > >
> > > Comment #1: now I am wondering as to what are the consequences of minimizing approximation error? Does it give an upper bound on total error accumulated?
> > >
> > > Response #1:
> > >
> > > Please read Response #2 first.
> > >
> > > (1) As mentioned in the Sec *INTRODUCTION*, the approximation error in value function results in *an overestimation phenomenon* and decreases the performance of RL algorithms ([1], [2], and [3]). It means that minimizing approximation error helps decrease the overestimation and increase performance.
> > >
> > > (2) We provided the results of the ablation study to verify the effect of KL constraint. We hope Reviewer #4 will run an check our code [https://github.com/SingerGao/ECAC] (**no any author information**)
> > >
> > > KL constraint==>decrease the approximation error==>increase performance.
> > >
> > > Comment #2: Can the authors elaborate on the point about connection to EM? Is this a formal connection?
> > >
> > > Response #2: When it is hard to **maximize** an objective (*log-likelihood for EM*) directly, one effective choice is to maximize a **lower bound** of the objective instead. Parallel, when it is hard to **minimize** an objective (*the approximation error for our method*) directly, one effective choice is to minimize an **upper bound** of the objective instead. This is why we derived an upper bound of the approximation error.
> > >
> > > Comment #3: I don't think that those prior works rely heavily on the KL-penalty -- any penalty that is similar to the KL should also work, so I am not sure about that part in the rebuttal.
> > >
> > > Response #3:
> > >
> > > We do not agree with this. [7] and [8] provide only an exact analysis of KL penalty rather than other regularizations.
> > >
> > > Comment #4: The auto-tuning the penalty was also done in SAC in the same way, so I am not sure why this is novel.
> > >
> > > Response #4:
> > >
> > > Please read our method carefully, our auto-tuning method is different from SAC.
> > >
> > > SAC: $\min_{\alpha}\mathbb{E}[-\alpha \log \pi(a_t|s_t)]-\alpha \bar H$.
> > >
> > > Our method: $\min_{\alpha}\mathop{E}\limits_{\boldsymbol{s}\sim P_\mathcal{D}}[\log \alpha \cdot ((\delta_{KL}+\delta_{entropy})-H].$
> > >
> > > Comment #5: It seems like Theorem 1 and Section 3 were removed (they are not there in the paper) anymore.
> > >
> > > Response #5: We removed Theorem 1 and Section 3 (in the initial submission) for the following reasons:
> > >
> > > - Sec 3 seems superfluous because Reviewer #1 and #4 excessively focused on it *but ignored the key idea of our work*: derive an upper boundary of approximation errors for the Q approximator and reduce the approximation error by minimizing this upper bound of the error.
> > > - This removal does not affect the main idea of our work.
> > >
> > > Comment #6: Having used a number of baselines myself, and from multiple repositories, it does seem to me that using the right set of baselines is needed ....
> > >
> > > Response #6:
> > >
> > > We used the official codebases of SAC (https://github.com/haarnoja/sac) and TD3 (https://github.com/sfujim/TD3) in our comparative evaluation. In our comparative evaluation, we used a set of random seeds different the official codebases, but all algorithms used the same set of random seeds. Have you ever tried to use different random seeds in the RL experiments? The performance of RL algorithms can be drastically different by using different random seeds [6].
> > >
> > > Comment #7: As I said in my review, there should also have been a theoretical/empirical discussion/comparison of alternate techniques like slower learning rate for the policy rather than an explicit KL constraint.
> > >
> > > Response #7:
> > >
> > > Slowing down the learning rate for the policy seems like the two-time-scale methods [4] and [5]. The idea of two-time-scale methods seems similar to the delayed policy update method (TD3 [3]) which only updates the policy after a fixed number of updates to the Q network. Delayed policy update is effective, but the benefits are limited. Moreover, *it may be hard to find a pair of two learning rates for all the tasks.* While the KL constraint stabilizes the Q function automatically. Besides, we do not provide the comparative evaluation about these in the revision because it needs time to compare the two-time-scale methods with the KL penalty.
> > >
> > > ## Reference
> > >
> > > [1] Sebastian Thrun and Anton Schwartz. Issues in using function approximation for reinforcement learning. 1993.
> > >
> > > [2] Philip Thomas. Bias in natural actor-critic algorithms. ICML 2014.
> > >
> > > [3] Scott Fujimoto, et al. Addressing function approximation error in actor-critic methods. ICML 2018.
> > >
> > > [4] Wu, Yue, et al. "A Finite-Time Analysis of Two Time-Scale Actor-Critic Methods." arXiv (2020).
> > >
> > > [5] Hong, Mingyi, et al. "A two-timescale framework for bilevel optimization: Complexity analysis and application to actor-critic." arXiv (2020).
> > >
> > > [6] Henderson, et al. Deep Reinforcement Learning that Matters. 2017.
> > >
> > > [7]  Nino Vieillard, et al. Leverage the average: an analysis of kl regularization in rl, 2020.
> > >
> > > [8] Matthieu Geist, et al. A theory of regularized Markov decision processes, 2019.

---

> > > > ### Comment · AnonReviewer1 · 2020-11-24
> > > > **Thanks for your response!**
> > > >
> > > > > EM connection
> > > >
> > > > For the EM part, I think a formal connection is needed. I see the rough analogy but we need a formal point there.
> > > >
> > > > > KL constraint==>decrease the approximation error==>increase performance.
> > > >
> > > > Is there a formal bound that shows this in the paper? I am happy to reconsider my evaluation if I missed this formal connection in the paper, the second part of this chain of arguments.
> > > >
> > > > > Auto-tuning SAC
> > > >
> > > > I think that's a minor difference, moreover, if you see the SAC implementation, they do actually use $\log \alpha$. I don't think I would call it too different.
> > > >
> > > > > Seeds, SAC, and TD3 baselines
> > > >
> > > > Thanks for your question, and yes, I have tried to use multiple random seeds. But, I guess the codebase for SAC is not the one you used, but https://github.com/rail-berkeley/softlearning. In fact on the repository that was used, it clearly redirects to this other repository.
> > > >
> > > > > Comparisons to learning rate changes
> > > >
> > > > I think there should be some. I see the point that it takes time to compare them, but I do think that any piece of research should have the right baselines.

---

### Official Review · AnonReviewer2 · 2020-11-02
**A more analysis on a known trick in reinforcement learning**

**Rating:** 6
**Confidence:** 4

**Review:**

Summary
=========
Authors investigated the effect of approximation error for actor-critic. They derived an upper bound of approximation showing that minimizing the KL divergence between the two consecutive policies can drive this upper bound down. Based on their finding they  introduced the Error Controlled Actor-critic (ECAC) algorithm. They ran ablation study showing the positive impact of minimizing the KL divergence. Furthermore they compared ECAC against 4 state-of-the-art techniques showing their advantage across 4 out of 5 Mujoco domains.

[+] Aside from minor grammar issues (see details) the paper is easy to follow

[+] The trick to calculate alpha and beta automatically is very interesting

[+] Empirical results are encouraging

[-] The experimental result section can enjoy more rigor. It is not clear how many seeds were used to generate these results. As Joelle discussed in one of NeurIPS conferences, it is easy use limited number of seeds and deduce very different outcomes (https://media.neurips.cc/Conferences/NIPS2018/Slides/jpineau-NeurIPS-dec18-fb.pdf). Also why showing min and max of returns instead of something more common such as std-err.

[-] I think Actor-critic has been shown to converge before. Why the proof using Banach’s Fixed Point Theorem important compared to other proofs?

Details
=========
P2, overestimation. => Need space after .
Section 2.2: hypothesis. And the => Remove "And"
Section 4.2:
	- that can be views adapting => that can be viewed as adapting
	- I recommend providing the difference between Q-networks and its target at n+1 iteration in a source distribution before the corresponding value on the target distribution (e.g. replace Eqn 11 and 12 and the accompanying text)
	- While the difference between D^n and D^n+1 is small, the corresponding error introduced can be big
	- I believe the use of minimized KL divergence between policies has been practiced before in the RL literature.
Section 4.3: ).In order => ). In order
Page 7:
	- Curves are smoothed uniformly for visual clarity" => What do you mean?
	- using different random seeds => How many random seeds? Why not mention here? I could not find it in the Appendix either
	- shaded region to the minimum and maximum returns over the five runs => Why not showing standard error instead to have a better statistical significance understanding visually?
	- Use vectorized image formats so your images are zoomable without the loss of quality
Eqn 23: Q_{ture} => Q_{true}
Page 8:
	- "with KL limitation have" => "with KL limitation has"
Section 5.2:
	- "ECAC outperforms all other algorithms on the tasks except" => What does beating mean? Have you ran statistical analysis or are you relying on the mean values?

---

> ### Author Response · Authors · 2020-11-22
> **Thanks for the reviewer's careful examination and good suggestions, and have improved clarity in the revised paper.**
>
> Thanks for the reviewer's careful examination and good suggestions. We have corrected the mistakes in our paper. Most importantly, to make our result analysis more rigorous, we add  Table 1 (Sec 4.2 in the revised paper) to show the results of max average return over five runs on all the 10 tasks.
>
> ## Responses to the comments
>
> ### comment #1:
>
> [-] The experimental result section can enjoy more rigor. It is not clear how many seeds were used to generate these results. As Joelle discussed in one of NeurIPS conferences, it is easy use limited number of seeds and deduce very different outcomes ([url]). Also why showing min and max of returns instead of something more common such as std-err.
>
> ### Response #1:
>
> (1)  We have updated our paper by adding a more clear statement about random seed in *Section EXPERRIMENTS* (Sec 4.2 in our revised paper): “Our results used five random seeds (each individual run applies a random seed) for 1) the Gym simulator, 2) network initialization, and 3) sampling actions from the policy during the training.”
>
> (2) Using the min and max of returns is also used frequently in RL papers (such as: [1], [2], and [3] in the following reference).  In order to monitor the stability of RL algorithms, the general approach is to evaluate the policy frequently, e.g. every 1, 000 time steps. Usually, each task is run for over 1 million time steps. Thus, usually, one algorithm on one task requires at least 1, 000 datapoints. If we plot the results by using error bar  (std-err), the plotting of the experimental results will be very crowded.
>
> (3) To make our result analysis more rigorous, we add  Table 1 (Sec 4.2 in the revised paper) to show the results of max average return over five runs on all the 10 tasks. Thank you very much for this suggestion!
>
> ### comment #2:
>
> [-] I think Actor-critic has been shown to converge before. Why the proof using Banach’s Fixed Point Theorem important compared to other proofs?
>
> ### response #2:
>
> Previous studies presented convergence proofs of actor-critic **with a certain approximation of value function**, e.g.  Degris et al. proved the convergence of the off-policy actor-critic **with a linear approximation of value function**.  While we must provide a convergence proof for actor-critic methods using the function approximation (**might be linear or non-linear approximation**).
>
> **We have removed the convergence analysis** (Section 3 in the first submission).  This removal does not affect the main idea of our work (i.e. minimize an upper bound of approximation error).
>
> ### comment #3:
>
> DetailsP2, overestimation. => Need space after . Section 2.2: hypothesis. And the => Remove "And" Section 4.2: - that can be views adapting => that can be viewed as adapting - I recommend providing the difference between Q-networks and its target at n+1 iteration in a source distribution before the corresponding value on the target distribution (e.g. replace Eqn 11 and 12 and the accompanying text) - While the difference between D^n and D^n+1 is small, the corresponding error introduced can be big - I believe the use of minimized KL divergence between policies has been practiced before in the RL literature. Section 4.3: ).In order => ). In order Page 7: - Curves are smoothed uniformly for visual clarity" => What do you mean? - using different random seeds => How many random seeds? Why not mention here? I could not find it in the Appendix either - shaded region to the minimum and maximum returns over the five runs => Why not showing standard error instead to have a better statistical significance understanding visually? - Use vectorized image formats so your images are zoomable without the loss of quality Eqn 23: Q_{ture} => Q_{true} Page 8: - "with KL limitation have" => "with KL limitation has" Section 5.2: - "ECAC outperforms all other algorithms on the tasks except" => What does beating mean? Have you ran statistical analysis or are you relying on the mean values?
>
> ### response #3:
>
> We appreciate the reviewer's careful examination. We have corrected the mistakes and improved clarity in the revision.
>
> ## Reference
>
> [1] John Schulman et al. Proximal policy optimization algorithms. *CoRR*, 2017.
>
> [2] Scott Fujimoto, Herke van Hoof, and David Meger. Addressing function approximation error in actor-critic methods. ICML 2018.
>
> [3] Tuomas Haarnoja, Aurick Zhou, Pieter Abbeel, and Sergey Levine. Soft actor-critic: Off-policy maximum entropy deep reinforcement learning with a stochastic actor. ICML 2018.
>
> [4] Thomas Degris, et al. Off-policy actor-critic, 2013.

---

### Author Response · Authors · 2020-11-23
**About revision.**

We appreciate all the reviewers, ACs, and PCs. The constructive comments and suggestions from the reviewers significantly improve the quality of our revised paper. We have corrected a number of mistakes and improved clarity in the revised paper. Moreover, the following are two main changes in the revision:

(1) We have removed the convergence analysis (Section 3 in the first submission). This removal does not affect the main idea of our work: derive an upper boundary of approximation errors for the Q approximator and reduce the approximation error by minimizing this upper bound of the error.  This idea is similar to the Expectation-Maximization Algorithm (EM) which maximizes a lower bound of the log-likelihood instead of the log-likelihood directly.

(2) To make our result analysis more rigorous, we add  Table 1 (sec 4.2 in the revision) to show the results of max average return over five runs on all the 10 tasks.


We have provided explanations for some unreasonable comments given by the reviewers.

---

### Decision · Program_Chairs · 2021-01-07
**Final Decision**

**Decision:**

Reject

**Comment:**

The majority of the reviewers believe that this paper is not ready for publication. Among their concerns is that the paper has limited novelty, especially in relation to existing work that use the KL constraint. Some of the reviewers also believe that the arguments are sometimes hand-wavy and not rigorous. For example, in the discussion period after Nov. 24th, it is mentioned that the argument by the authors that "KL constraint==>decrease the approximation error==>increase performance" is not precise enough. I encourage the authors to take these comments into account and improve their paper.